

# Predicting repeat power ability through common field assessments: is repeat power ability a unique physical quality?

Alex O. Natera[1,2], Dale W. Chapman[3], Neil D. Chapman[1] and Justin W.L. Keogh[1,4,5]

[1] Faculty of Health Sciences and Medicine, Bond University, Gold Coast, Queensland, Australia
[2] Sport Science, New South Wales Institute of Sport, Sydney Olympic Park, New South Wales, Australia
[3] Curtin School of Allied Health, Curtin University, Perth, Western Australia, Australia
[4] Sports Performance Research Centre New Zealand, Auckland University of Technology, Auckland, New Zealand
[5] Kasturba Medical College, Manipal Academy of Higher Education, Manipal, Karnataka, India

Corresponding author
Alex O. Natera,
alex.natera@hotmail.com

## ABSTRACT

**Background**. The repeat power ability (RPA) assessment is used to test the ability to repeatedly produce maximal ballistic efforts with an external load. The underpinning physical qualities influencing RPA are undetermined. This study aimed to gain further insight into the physical qualities that determine RPA by analysing the association between physical qualities and an assessment of RPA.

**Materials and methods**. Ten well-trained male field hockey players performed an RPA assessment consisting of 20 repetitions of loaded countermovement jumps (LCMJ20), with a percent decrement score of peak power output calculated. Over a two-week period, each participant performed the YoYo Intermittent Recovery Test 2 (IRT2), a repeated speed ability assessment incorporating a 180° change of direction ($RSA_{180}$), a 40-meter linear speed test (40 mST), an isometric mid-thigh pull (IMTP), a countermovement jump (CMJ), and a 3-repetition maximum half squat (HS) assessment. Pearson's correlation analysis was used to determine the strength of relationships between each assessment variable and the LCMJ20. The assessment variables with the strongest relationships within each assessment were used in a stepwise multiple linear regression analysis to determine the best predictor model of LCMJ20.

**Results**. $RSA_{180}$ percent decrement score ($RSA_{180}$% had a very strong, significant relationship with LCMJ20 ($r = 0.736$: $p < 0.05$). HS relative strength (HSrel) was found to have a significant and very strong, negative relationship with LCMJ20 ($r = -0.728$: $p < 0.05$). Stepwise multiple linear regression analysis showed RSA180 to explain 48.4% of LCMJ20 variance (Adjusted $R^2 = 0.484$) as the only covariate included in the model.

**Conclusion**. The findings indicate that $RSA_{180}$ as a repeated high intensity effort (RHIE) task is strongly related to LCMJ20 and is also the best predictor of LCMJ20. This may suggest that RPA can provide practitioners with information on RHIE performance. The variance between assessment methods indicates that RPA may be a distinct physical quality, future research should assess other physical capacities to better understand the factors contributing to RPA.

# INTRODUCTION

Repeat power ability (RPA) can be described as the ability to repeatedly produce maximal or near maximal efforts against external resistance training loads (*Natera, Cardinale & Keogh, 2020*). The ability to maintain high levels of maximal power output is suggested to be important for performing repeated high intensity efforts (RHIE) in many sports (*Apanukul, Suwannathada & Intiraporn, 2015*; *Gabbett & Wheeler, 2015*; *Gonzalo-Skok et al., 2016*; *Goods et al., 2022*; *Natera, Cardinale & Keogh, 2020*; *Schuster et al., 2018*; *Vachon et al., 2021*). As such, there may be a concomitant need to effectively monitor training induced changes in RPA (*Apanukul, Suwannathada & Intiraporn, 2015*; *Baker & Newton, 2007*; *Natera, Cardinale & Keogh, 2020*; *Schuster et al., 2018*). To better understand and define RPA, and to determine appropriate training modalities that may be used to enhance RPA, a thorough analysis of associated physical qualities needs to be undertaken. The associations between neuromuscular qualities like maximal strength, maximal power output and speed are important to understand so training to enhance RPA can be emphasised and monitored accordingly (*Baker & Newton, 2007*; *Mosey, 2011*; *Natera, Cardinale & Keogh, 2020*). Similarly, the association between cardiorespiratory performance like aerobic capacity and repeat speed ability are also important to clearly understand RPA and the underpinning cardiorespiratory qualities that may determine RPA performance (*Fry et al., 2014*; *Natera, Cardinale & Keogh, 2020*; *Sands et al., 2004*).

RHIEs is a cluster of three or more high intensity sporting actions, like accelerating, cutting, tackling, and jumping with a maximum of 21 s rest between each high intensity effort (*Spencer et al., 2004*). RHIEs have been found to occur around critical periods of play and often lead to successful outcomes in team sports (*Black & Gabbett, 2015*; *Gabbett & Gahan, 2016*; *Hulin et al., 2015*). The ability to perform RHIEs is suggested to reflect the ability to maintain relatively high levels of maximal power output under some level of fatigue (*Natera, Cardinale & Keogh, 2020*). RPA assessments may provide useful information to determine RHIE performance. Despite the potential importance of RPA, there is little known about the underpinning physical qualities that are associated with or determine RPA.

Although direct investigations to determine RPA have not been conducted, several studies have looked at the relationships between anaerobic performance, as measured *via* a 30 s Wingate assessment, and various forms of repeated jump and repeated speed squat performance (*Bosco, Luhtanen & Komi, 1983*; *Sands et al., 2004*; *Fry et al., 2014*). Strong relationships have been reported between these RPA assessments, consisting of either continuous compliant rebound jumps or speed squats, and the 30 s Wingate assessment conducted on cycle ergometry (*Bosco, Luhtanen & Komi, 1983*; *Sands et al., 2004*; *Fry et al., 2014*). Despite the mechanical differences between these assessments, the strong

relationship identified is likely due to both assessments requiring maximal intensity effort of the lower limbs over a similar duration of approximately 30 s.

Although several RPA assessments have been described in the literature, an assessment utilising 20 loaded counter movement jumps (LCMJ20) has recently been reported as a reliable RPA assessment that may better replicate sporting movements and can be easily administered in the field (*Natera, Cardinale & Keogh, 2020*; *Natera et al., 2023*). The LCMJ20 may be considered the most effective and reliable assessment of RPA, as the researchers examined a range of different jump types, measurement indices, calculations and methods of describing power decline in their study. The most reliable method to quantify RPA is to calculate peak power output for each jump of the LCMJ20 and then to calculate a percent decrement score (percent decrement $= 100 \times$ [total jump power/ideal jump power]$-100$) with the first and last jumps removed from the calculation (*Natera et al., 2023*).

With the current lack of research and understanding of the physical qualities that contribute to RPA, there is a need for further investigations. A greater understanding of the potential contributing physical qualities, may help practitioners design training programs to enhance RPA more specifically. On the other hand, RPA might lack any associations with common assessed physical qualities and this may suggest that RPA is somewhat unique and may require specific training in order to develop it.

The purpose of this research is to identify the underlying physical qualities that determine RPA. This study seeks to provide the practitioner and coach with a better understanding of RPA and the potential to target training towards the development of physical qualities that can best enhance RPA. We hypothesise that RPA will have substantial unexplained variance and that strong relationships will exist between RPA and RHIE performance.

## MATERIALS & METHODS

### Study design

A cross-sectional study design was used to investigate a range of field test assessments commonly used in team sports. This research was designed to align with the assessments and timing of assessments commonly used by practitioners working in team sport. Furthermore, the demands of the preseason training plan for the field hockey players participating in this study was accounted for. All participants were familiar with the assessments used in this study as they were routine assessments used by this particular cohort.

Following a further session of familiarization with the LCMJ20, participants were required to attend the strength and conditioning facility on four different occasions over a two-week period, two sessions in week one and two sessions in week two, with all sessions performed at the same time of day (Fig. 1). In session one, descriptive data was collected and countermovement jump (CMJ), isometric mid-thigh pull (IMTP) and half squats (HS) strength were assessed respectively with 20 min of rest between each assessment. During session two, which was conducted 72 h after session one, the LCMJ20 RPA assessment was performed. In the following week, session three included a 40 meter sprint (40 mST) and a repeated sprint assessment with a change of direction (RSA$_{180}$). Between each of
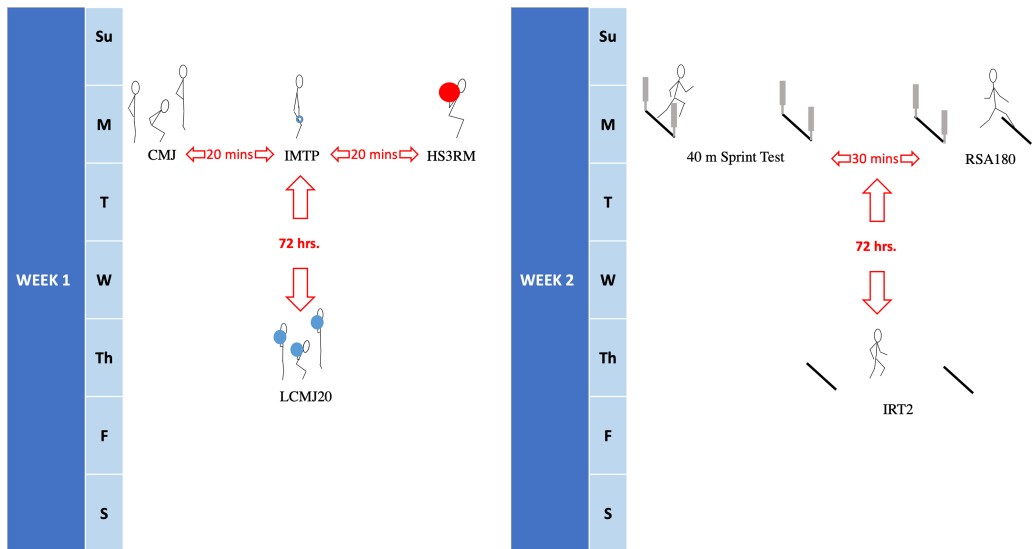

**Figure 1  Data collection procedure.** The 2 week data collection period, indicating week 1 and week 2 assessments. Assessments on week 1, Monday: CMJ (countermovement jump), IMTP (isometric mid-thigh pull) and HS3RM (half squat 3RM). Assessment on week 1, Thursday: LCMJ20 (20 loaded countermovement jumps). Assessments on week 2, Monday: 40 m sprint test and RSA180 (repeat speed ability assessment with change of direction). Assessment on week 2, Thursday: IRT2 (Yoyo intermittent recovery test 2).

these assessments, 30 min of recovery was provided. The Yoyo intermittent recovery test 2 (IRT2) was then conducted 72 h later in session four.

All of the running based assessments used in this study (40 mST, $RSA_{180}$ and IRT2) matched the protocols and requirements of the National Sporting Organisation, Hockey Australia and fulfilled part of their biannual player assessment batteries. As such all running assessments in this study were conducted on the same indoor tartan surface and all participants were familiar with the assessments used in this study.

Technical hockey training during the two week assessment period was modified to be low intensity, skill based training to facilitate recovery between sessions and to optimise both neuromuscular and cardiorespiratory performance for the assessments. A standardised warm up was conducted prior to the first assessment on each testing day. The warm up consisted of 5 min of cycling at a rate of perceived exertion of 3–4, followed by 2 min of dynamic flexibility exercises for the lower limb and trunk.

## Participants

Ten well-trained male field hockey players currently playing at state level competition volunteered to participate in this study (age = 22.30 ± 2.41, body mass = 79.4 ± 5.6 kg, stature 182.3 ± 5.5 cm). The sample size was affected due to local COVID-19 pandemic restrictions. This was a sample of convenience where all participants met the inclusion criteria of a minimum of three years resistance training experience, regularly trained with heavy squats and loaded jumps, and able to travel to the testing facility without breaking local COVID-19 travel restrictions.

Assessments were conducted during pre-season after six weeks of incremental training involving hockey specific drills, speed and agility training and a range of intensities used in interval training. Prior to commencing the study, a full explanation of the assessment procedures was given to each participant including a training history questionnaire, to verify their study eligibility. Following an opportunity to ask any questions all participants provided their written informed consent. The investigation was approved by Bond University Human Research Ethics Committee (N00156). Participants who were currently injured, or had not consistently trained over the previous 6 weeks were excluded from this study.

All participants were required to consume a standardised meal approximately 2 h before each testing session and only consume water during the testing sessions. Each pre-session meal consisted of carbohydrates ($1$–$1.5$ g kg$^{-1}$), protein ($0.3$ g kg$^{-1}$) and fat ($0.28$–$0.47$ g kg$^{-1}$). All participants were instructed not to consume coffee or caffeine products on the assessment days and to maintain their normal nutrition practices on all other non-testing days. The pre-assessment consumption was confirmed with each participant prior to commencing the warm up.

## Countermovement jump

A dual-force plate system (ForceDecks; Vald Performance Systems, Brisbane, Australia) sampling at 1,000 Hz was used to collect force-time data for the counter movement jump (CMJ). The CMJ assessment consisted of five individual jumps performed on the force plates after a specific warm up consisting of three submaximal jumps at 50, 75 and 90% maximal effort (*Petrigna et al., 2019*). At the completion of the specific warm up, 2 min of recovery was provided before the CMJ assessment was performed. Each jump in the CMJ assessment was separated by 30 s and each participant was required to stand motionless for up to 5 s before initiating the downward phase of the jump (*Warr et al., 2020*). The participants were instructed to keep their hands on their hips, use their own self-selected depth and execute the jump "pushing off the ground as hard and as fast as possible" whilst trying to reach a maximal height (*Hughes, Peiffer & Scott, 2020*). A self-selected countermovement depth was chosen to allow each participant to naturally express their preferred jump strategy in order to optimise jump performance (*Jidovtseff et al., 2014*; *Mandic, Jakovljevic & Jaric, 2015*; *McBride et al., 2010*). Discrete jump data was extrapolated off the automated software with the jump registering the greatest peak power output, absolute and relative, and the jump registering the best height used for further analysis. Reliability for CMJ height, CMJ absolute peak power output and CMJ relative peak power output were, ICC = 0.97, 0.97 and 0.93, and CV = 1.9, 1.8, and 1.8 respectively.

## Isometric midthigh pull

Following a specific IMTP warm up consisting of $3 \times 3$ s efforts at 50, 75 and 90% of maximal effort with 60 s rest, a maximal IMTP assessment was performed. A minimum of $2 \times 3$ s maximal effort trials were performed on the force plates within a customised IMTP testing rig using the same force plate system and software (ForceDecks; Vald Performance Systems, Brisbane, Australia) used in the CMJ assessment. If the difference between trials was >250 N a third trial was performed (*Comfort et al., 2019*).

The bar was positioned so that when each participant gripped the bar and positioned their feet hip-width under the bar, the bar contacted their thighs close to the inguinal crease. While using lifting straps and athletic tape, each participants' hands were secured to the bar with an overhand grip with knees at an angle between 125–145° and hips at an angle between 140–150° (*Comfort et al., 2019*). An upright torso with shoulders above or slightly behind the vertical plane of the bar was also established.

The participants were instructed to "to push your feet into the ground as hard and as fast as possible" and after the command "3, 2, 1, PULL!", the participants performed the IMTP assessment. After each trial, the force trace was visually monitored to ensure that body weight was not >50 N and that there was a clear stable period and no obvious countermovement prior to initiating the pull. Peak force was extrapolated from the software and the best absolute and relative peak force scores were used for further analysis. IMTP reliability for absolute ($IMTP_a$) and relative peak force ($IMTP_{rel}$) was established (ICC = 0.78 and 0.81; CV = 4.8 and 4.8 respectively).

### Estimated one repetition maximum half squat

The 3RM HS assessment was conducted on a Smith machine with a full range of weight plates. A high bar placement was used where the bar was placed on the upper trapezius muscles, and squat descent was monitored to a knee angle of 90°. The descent depth was established with the use of a goniometer and also visually monitored. As a further precaution, a thin rubber band was positioned so that the posterior thighs of each participant would contact the band when the required depth was reached (*Thomasson & Comfort, 2012*). Video recording from the sagittal plane was further used to monitor the squat depth. The use of a Smith machine and the control of squat depth were required in order to match the equipment and depth of jump performed in the RPA assessment.

Over three to four sub-maximal sets of one repetition, each participant gradually built to their estimated 3RM load (*Cronin & Hansen, 2005*). The participants then attempted their estimated 3RM and with every successful attempt, after three minutes of passive recovery, 5 kg was added until their 3RM was reached (*Cronin & Hansen, 2005*). Following the procedures of *Natera et al. (2023)*, the last successful 3RM HS load lifted was converted into an estimated 1RM using the average of seven different 1RM estimation formula's (*Brzycki, 1993*; *Epley, 1985*; *Lander, 1985*; *Lombardi, 1989*; *Mayhew et al., 1992*; *O'Connor & S, 1989*; *Wathen, 1994*).

The validity of 1RM calculations can be influenced by the exercise performed and equipment used, eg. back squat *versus* leg press and free bar *versus* Smith machine (*Hughes, Peiffer & Scott, 2020*; *LeSuer et al., 1997*; *Wood, Maddalozzo & Harter, 2002*) and the population assessed, eg. gender, age and training history (*Bianco et al., 2015*; *Mayhew et al., 2008*; *Ritti-Dias et al., 2011*). It was considered a safer option to follow the procedures of *Natera et al. (2023)* and use an average of these different 1RM estimations. The mean estimated 1RM derived from these equations was used for further analysis as both an absolute ($HS_{ab}$) and relative to body mass load ($HS_{rel}$).

## Repeat power ability assessment

The RPA assessment consisted of 20 maximal repetitions of loaded CMJs with a predicted 30% 1RM load, extrapolated from the 3RM HS assessment (*Natera et al., 2023*). The same dual-force plate system (ForceDecks; Vald Performance Systems, Brisbane, Australia) sampling at 1,000 Hz was used to collect force-time data for all repetitions of the LCMJ20. Five minutes prior to commencing the LCMJ20 assessment, a specific warm up consisting of 3 LCMJs with the 30% 1RM load were performed. Each LCMJ was separated by 10 s and was performed at a perceived intensity of 50%, 75% and 100%, respectively. Peak power output for the final warm up LCMJ was collected with the linear position transducer and used as a "power standard" for the LCMJ20.

Each jump in the LCMJ20 was performed every 3 s, using a metronome to maintain the rhythm at a tempo of 20 beats per minute. The timing of each jump provided limited inter-repetition rest (∼1.5–2 s) and allowed for precise data collection. A Smith machine was used, with the fixed barbell held in the high bar position, with the depth of each jump visually and kinesthetically monitored to a depth of 90° knee flexion (using a thin elastic band). Upon landing from each jump, participants assumed an upright posture with knees and hip extended in stance until the sound of the metronome. The sound of the metronome signaled the immediate initiation of the countermovement of the next jump.

Whilst the force plates were used to collect force-time data, a linear position transducer was also used to provide derived instantaneous feedback of peak power output for each jump. A tablet screen was positioned directly in front of each participant to provide visual feedback. Along with the visual feedback, real time verbal feedback of power output was provided by the researchers. The researchers also provided ongoing verbal motivation to each participant to perform every jump at maximal intent. Within the first three jumps of the RPA assessment each participant was required to reach their "power standard" in order for the assessment to continue. No data from the linear position transducer was used for further analysis.

In following the procedures established by *Natera et al. (2023)*, customised Matlab code was used to derive peak power output for jumps 2 to 18 from the force-time data. In order to reduce drift caused by integrating derived acceleration data, acceleration data was filtered using a second-order high-pass Butterworth filter. A cutoff frequency of 0.25 Hz was used to filter the acceleration data. The following formula was used to calculate a percent decrement score which was used in further analysis as the RPA value for the LCMJ20: Fatigue $= 100 \times$ [total jump power/ideal jump power]$-100$.

## Speed

Prior to the linear speed test, a specific warm up consisting of three incremental sprints over 20, 30 and 40 m at a perceived intent of 70, 80 and 90% intensity, respectively were performed. After 5 min rest, $5 \times 40$ m sprints were performed with 3 min recovery between each sprint. Dual beam timing gates (Smartspeed Pro; Fusion Sport, Queensland, Australia) positioned at the start and 40 m line were used to capture sprint times with each participant starting in a two-point stance with their lead foot exactly 0.5 m behind the start

line. Reliability was established (ICC = 0.93; CV = 3.4) and each participant's fastest 40 m sprint (40 mST) was used for further analysis.

### Repeated Speed Ability

The repeated speed test consisted of 6 × 40 m sprints with a 180° change of direction at the 20 m mark (20 m + 20 m with a 180° turn) (*Goods et al., 2022*). At the start of the first sprint a timer was started, and each consecutive sprint commenced every time a 30 s period had elapsed. Each participant started from a static position in a two-point stance, with their lead foot exactly 0.5 m behind dual beam timing gates (Smartspeed Pro; Fusion Sport, Queensland, Australia). Each participant was required to touch the 20 m line with their foot before performing the 180° change of direction and sprinting back to the start line and through the timing gates. Foot placement on the 20 m line was video recorded to ensure all trials were valid.

After each sprint, the participants decelerated over a marked 10 m zone and walked back to the start line to await the next sprint. A 5 s countdown was provided as timing for each following sprint. A percent decrement score was used to quantify speed decline with the following equation Fatigue = 100 × [total sprint time/ideal total sprint time]−100.

### Yoyo intermittent recovery test 2

The IRT2 consisted of 2 × 20 m shuttle runs performed at increasing speeds. The timing of each shuttle was controlled by recorded audio beeps played over a portable speaker system. Each shuttle run was interspersed with 10 s of recovery. If a participant failed to reach the start line at the sound of a beep, a warning was given. If for a second time the participant failed to reach the start line at the sound of a beep, the last successful shuttle was recorded as their test result. Total distance achieved in the IRT2 (IRT2m) was used for further analysis. Reliability for the IRT2m has previously been established in male team sport players with an ICC of 0.96 and a CV of 4.2% (*Enright et al., 2018*).

### Statistical analysis

Statistical analysis was conducted using JASP online statistical analysis package (Version 0.16.4; *JASP Team, 2022*). Data centrality and spread for continuous variables is reported as means and standard deviations with data checked for normality of distribution using the Shapiro–Wilk test. Pearson's correlations were used to identify relationships between LCMJ20 and the physical qualities tested. The thresholds used to interpret the correlation magnitude were: trivial < 0.1; small, 0.11–0.3; moderate, 0.31–0.5; large, 0.51–0.7; very large, 0.71–0.9 and almost perfect, 0.91–1.0 (*Hopkins, 2010*).

In a secondary analysis a stepwise regression analysis was conducted in order to determine which combination of outcome variables (CMJcm, $IMTP_{rel}$, $HS_{rel}$, $RSA_{180}$%, IRT2m) established the strongest possible prediction model for the LCMJ20. Only variables identified with a moderate or better Pearson's correlation were used as predictor variables in stepwise regression analysis to establish the best single predictor model of LCMJ20. Statistical significance was accepted at $p < 0.05$.

**Table 1 Descriptive statistics for all variables used in correlation analysis.** Data for each variable presented in means ± standard deviations.

| Assessment variable | Mean ± SD |
| --- | --- |
| CMJ relative power (W/kg) | 57.45 ± 3.06 |
| IMTP relative peak force (N/N) | 4.32 ± 0.38 |
| Half squat predicted 1RM (kg) | 183.4 ± 17.1 |
| LCMJ20 (%) | 26.81 ± 3.8 |
| 40 m sprint time (s) | 5.25 ± 0.23 |
| RSA decrement (%) | 2.85 ± 1.1 |
| IRT2 distance (m) | 820 ± 238.7 |

**Notes.**

CMJ, countermovement jump; IMTP, isometric midthigh pull; 1RM, predicted 1 repetition maximum; peak power output percent decrement score of 20 repetitions of loaded countermovement jumps; repeated speed ability assessment with 180 degree change of direction; IRT2, yoyo intermittent recovery test 2.

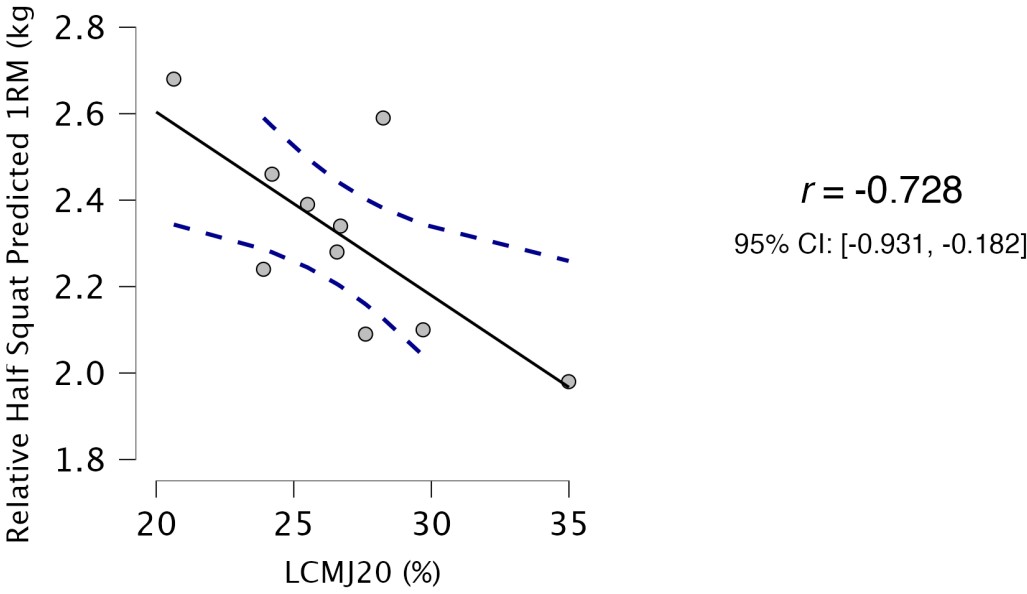

**Figure 2 Relationship between LCMJ20 and relative HS_rel.** Full black line indicating a strong negative relationship and blue dashed lines indicating 95% confidence intervals. LCMJ20: peak power output percent decrement score of 20 repetitions of loaded countermovement jumps; 1RM: 1-repetition maximum.

## RESULTS

Descriptive statistics for all variables used in correlation and regression analysis are presented in Table 1. Both HS_rel (Fig. 2) and RSA_180 (Fig. 3) had very strong, significant relationship with LCMJ20 ($r = -0.728$, $p = 0.017$ and $r = 0.736$, $p = 0.015$ respectively). All other variables exhibited trivial to weak non-significant relationships with LCMJ20 (ranging from $r = -0.293$ to $0.118$) (Table 2).

Percent decrement in speed for RSA_180 was the only identified covariate in the predictor model of LCMJ20 with stepwise multiple regression analysis predicting 48.4% of the variance in LCMJ20, Adjusted $R^2 = 0.484$, $F(1,8) = 9.43$, $p = .015$.

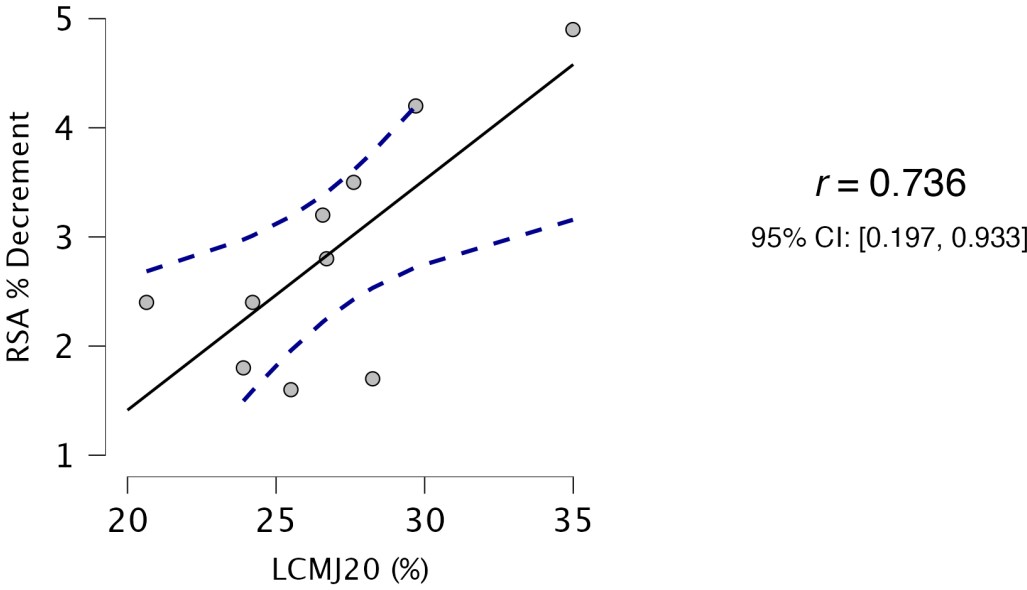

**Figure 3  Relationship between LCMJ20 and RSA₁₈₀.** Full black line indicating strong positive relationship and blue dashed lines indicating 95% confidence intervals. LCMJ20: peak power output percent decrement score of 20 repetitions of loaded countermovement jumps; RSA: repeated speed ability assessment.

**Table 2  Pearson's correlations between LCMJ20 and other physical fitness test variables.**

| Field test variables | Pearson's r | p | Lower 95% CI | Upper 95% CI |
|---|---|---|---|---|
| Relative HS 1RM (kg/kg) | −0.728 | 0.017[*] | −0.931 | −0.182 |
| IRT2 distance (m) | −0.27 | 0.45 | −0.769 | 0.433 |
| RSA % decrement (%) | 0.736 | 0.015[*] | 0.197 | 0.933 |
| Best 40 m sprint time (s) | 0.118 | 0.745 | −0.553 | 0.696 |
| IMTP relative peak force (N/N) | −0.204 | 0.572 | −0.739 | 0.488 |
| CMJ relative power (W/kg) | −0.293 | 0.411 | −0.779 | 0.413 |

**Notes.**

[*]$p < 0.05$.

HS 1RM, predicted half squat 1-repetition maximum; IRT2, yoyo intermittent recovery test 2; RSA, repeated speed ability assessment; IMTP, isometric midthigh pull; CMJ, countermovement jump.

# DISCUSSION

The aims of this study were to identify the underlying physical qualities that determine RPA. With a better understanding of RPA, practitioners wishing to develop RPA can better develop their exercise prescription to develop it more effectively.

The results of this study demonstrate that approximately half of the variance of RPA can be explained by IRT2, RSA₁₈₀, 40mST, IMTP$_{rel}$, CMJ and HS$_{rel}$. More specifically, the RPA variance explained is largely determined by RSA₁₈₀ percent decrement. The prediction of RPA only slightly improved with the addition of HS$_{rel}$. Overall, these results suggest there is substantial variance in RPA left unexplained when only using these selected field assessments.

The RSA$_{180}$ can be described as a RHIE assessment, with four high intensity actions performed over a brief period ($\sim$7 s). The deceleration and change of direction actions in repeated shuttle performance presents a higher physiological load than that experienced in repeated effort straight line sprinting (*Buchheit et al., 2010*). The physiological and biomechanical demands between the RSA$_{180}$ and the LCMJ20 may have similarities based on the intensity of effort, the total duration of effort and the mechanical demands of both tasks. Specifically, the RSA$_{180}$ and the LCMJ20 are maximal intensity assessments that require high braking and propulsive forces to execute both the deceleration and acceleration and landing and propulsive elements of each assessment, respectively (*Dos'Santos et al., 2017*; *Lake et al., 2021*).

The stretch shortening cycle and corresponding elastic contribution of the muscle tendon units of the lower limb would also appear important for both assessments (*Seiberl et al., 2021*). Total duration of maximal effort is likely to be similar between both assessments. Total time of effort in the RSA$_{180}$ was 42.3 $\pm$ 1.5 s and when accounting for the braking, propulsive and landing phases of each jump, total time in the LCMJ20 is likely to be $\sim$35 s, with time per jump ranging from 1.5 s in the earlier repetitions (*Lake et al., 2021*; *Pérez-Castilla et al., 2021*) and, in visually inspecting the force-time data in this study, up to $\sim$2 s in later repetitions.

Despite the cyclical action of running and the acyclical action of the loaded CMJs, very strong relationships exist between the assessments due to these physiological and biomechanical similarities between the tasks. The strong association between the RSA$_{180}$ and the LCMJ20 RPA assessment may suggest that LCMJ20 is a good assessment of the underpinning physical quality(s) for high level RHIE performance.

HS$_{rel}$ had a very strong negative relationship with LCMJ20, indicating that higher relative squat performance is associated with lower power decrement and therefore better RPA performance. As predicted 1RM loads were used to quantify external loads for the LCMJ20, participants with higher relative squat performance used higher absolute loads during the LCMJ20. Greater levels of HS strength improvement have previously been shown to enhance repeated high intensity performance (*Bogdanis et al., 2011*). Participants with greater HS strength performed better in the second half of a repeated sprint cycle assessment consisting of 10 $\times$ 6 s sprints interspersed with 24 s of passive recovery (*Bogdanis et al., 2011*). A similar finding can be observed in the current study with a strong negative relationship found between HS$_{rel}$ and RSA$_{180}$.

It is suggested that stronger athletes may have better fatigue resistance through improved recruitment patterns and the ability to increase motor unit activity at the end of a repeated sprint test (*Bogdanis et al., 2011*). Similar neural adaptations may influence fatigue resistance in the LCMJ20. Additionally, both the externally applied load and the similar movement patterns used in the HS and the LCMJ20 are also likely to influence the very strong associations between the measures in comparison to the other assessments used in this study. Despite the strong associations between HS$_{rel}$ and LCMJ20, the importance of HS$_{rel}$ in predicting RPA performance is questionable. The inclusion of HS$_{rel}$ to the predictive model only marginally improved the model. The lack of effect of HS$_{rel}$ on the model may be due to the differences in the duration of both activities.

The inclusion of IRT2m in the predictive model also made little difference; with the model's predictive ability slightly decreasing. The IRT2 is an intermittent field test with strong relationships to aerobic performance (*Bangsbo, Iaia & Krustrup, 2008*). The aerobic system has been found to be a key component of recovery between repeated sprints with the replenishment of phosphocreatine driven by aerobic processes (*Turner & Stewart, 2013*). Unlike repeated sprint performance, the LCMJ20 performance and the ability to repeat power does not seem to be highly determined by aerobic processes. This may reflect the short duration of each jump and recovery period being too brief for the aerobic system to influence recovery. In comparison to the IRT2, the intensity of effort in the LCMJ20 is maximal with a relatively short duration of effort. Anaerobic processes would therefore largely determine LCMJ20 performance, and this is evident in the strong relationship found between the Wingate assessment and other RPA assessments, with relationship ranging between $r = 0.70$–$0.89$ (*Bosco, Luhtanen & Komi, 1983*; *Sands et al., 2004*; *Pupo et al., 2013*; *Fry et al., 2014*).

Despite the very strong relationship found between $RSA_{180}$ and LCMJ20, there is substantial variance still left unexplained. With previous research identifying strong relationships between Wingate cycling performance and RPA assessments (*Bosco, Luhtanen & Komi, 1983*; *Sands et al., 2004*; *Fry et al., 2014*), a more direct measure of anaerobic capacity may have improved the predictive model in this study. However, in this study cardiorespiratory characteristics were examined using mechanically matched field assessments, all requiring running and the concomitant use of the stretch-shortening cycle. The LCMJ20 differs to several other RPA assessments reported in the literature (*Bosco, Luhtanen & Komi, 1983*; *Sands et al., 2004*; *Fry et al., 2014*), as it requires the use of the stretch-shortening cycle, a greater requirement for acceleration throughout the propulsive part of the movement and as a result of using external load, an increased mechanical demand placed on the propulsive and landing phases of the jump (*Lake et al., 2021*; *Natera et al., 2023*). The effect of additional load on landing phase force-time characteristics in the LCMJ20 is likely to further differentiate the Wingate assessment from the LCMJ20.

Unlike $HS_{rel}$, maximal outputs of lower limb force, lower limb power and speed did not improve the predictive model. These findings suggest that RPA, as measured by the LCMJ20, is not determined by maximal neuromuscular outputs. It is important to note that the force-time analysis used in the current study relies on discrete variables, taken from a single point in a jump, to explain jump performance. With the LCMJ20 used to assess the magnitude of decline in RPA performance, a more nuanced analysis methods to capture, detect and describe the differences in the whole force-time trace may be required to better describe RPA performance.

Although this study sought to describe RPA through commonly used field assessments, a limitation of this study was to have not included more sophisticated measures of both neuromuscular and cardiorespiratory function to help better explain and predict RPA. For example, surface electromyography may have been used to better understand changes in myoelectrical output of the lower limb muscles during the LCMJ20 (*Felici & Del Vecchio, 2020*). Near-infrared spectroscopy could also have been used to better understand various physiological and metabolic factors associated with LCMJ20 performance

(*Tuesta et al., 2022*). Metabolic cart derived metrics, that are commonly utilised with aerobic and anaerobic ergometry based assessments, may have also provided additional variables that could have improved the predictive ability of our analyses. Considering the relationships between the 30 s Wingate assessments and other RPA assessments, the non-use of the Wingate assessment in this study may also be considered a limitation. Lastly, the sample size used in this study should be considered in any interpretation of the results.

Futures lines of investigations may look to include more sophisticated measures of neuromuscular and cardiorespiratory function to attempt to explain more of the variance in RPA and ultimately provide a more complete understanding of the factors contributing to RPA. Further investigations may also look to apply a more nuanced analysis of the whole LCMJ20 force-time waveform to gain greater insight into what exact changes occur across the 20 repeated CMJs. Lastly, further investigations may also look to develop RPA and assess its influence on RHIE performance through specific training methods like high volume power training (HVPT) (*Apanukul, Suwannathada & Intiraporn, 2015*; *Gonzalo-Skok et al., 2016*; *Natera, Cardinale & Keogh, 2020*; *Schuster et al., 2018*).

Despite the limitations of this study the findings suggest that RPA is in part a unique physical quality with substantial unexplained variance from the field assessments used. With $RSA_{180}$ having the strongest associations, and the only variable in the Stepwise predictor model to explain LCMJ20, RPA may well be an important physical quality that underpins RHIE performance. However, interpretation of the results of this study should be taken with care when applying to different participant groups or athletic cohorts. It is important to acknowledge that team sport athletes such as the field hockey players in this study, require a good combination of neuromuscular and cardiorespiratory development whereas a sprinter requires a highly developed neuromuscular system and an endurance athlete a highly developed cardiorespiratory system (*Thompson, 2017*; *Lombard et al., 2021*). Different findings to the current study might occur with athletes with more highly developed physical characteristics unique to their sports.

## CONCLUSION

In attempting to identify the underlying neuromuscular and cardiorespiratory qualities that determine RPA, we found that approximately half of the variance in RPA can be explained by the common field assessments used in this study. With RPA having strong associations and strong predictive ability by $RSA_{180}$, we propose that RPA may be an underpinning physical quality that can be used to partly determine and assess RHIE performance. Other field assessments used in this study did not seem to associate with RPA performance and this may further highlight the potential uniqueness of RPA as a physical quality.

The LCMJ20 is an assessment of RPA that can be used in the field by practitioners and coaches to provide insight into the ability of their athletes to maintain maximal power output. With RHIEs having a strong relationship and predictive ability of RPA, the development of RPA may improve RHIEs in team sports like field hockey. Results of this study suggest that RPA is a somewhat unique physical quality that can underpin RHIE performance but is not associated with other common field-assessed physical qualities

of fitness and strength. Practitioners can assess RPA and then determine whether to use specific training methods like HVPT to develop and enhance RPA and RHIE performance.

### Funding
The authors received no funding for this work.

### Competing Interests
Justin Keogh is an Academic Editor for PeerJ.

### Author Contributions
- Alex O. Natera conceived and designed the experiments, performed the experiments, analyzed the data, prepared figures and/or tables, authored or reviewed drafts of the article, and approved the final draft.
- Dale W. Chapman conceived and designed the experiments, authored or reviewed drafts of the article, and approved the final draft.
- Neil D. Chapman conceived and designed the experiments, authored or reviewed drafts of the article, and approved the final draft.
- Justin W.L. Keogh conceived and designed the experiments, authored or reviewed drafts of the article, and approved the final draft.

### Human Ethics
The following information was supplied relating to ethical approvals (i.e., approving body and any reference numbers):

Bond University Human Research Ethics Committee granted ethical approval to carry out this study.

### Data Availability
The data is available at Figshare: Natera, Alex (2023). Raw force-time data for Loaded CMJs. figshare. Dataset. https://doi.org/10.6084/m9.figshare.24069861.v1

Natera, Alex (2023). Determinants Data - deidentifed.csv. figshare. Dataset. https://doi.org/10.6084/m9.figshare.24069837.v1

The code is available at Figshare: Natera, Alex (2023). Matlab code for LCMJ20 data. figshare. Dataset. https://doi.org/10.6084/m9.figshare.24072720.v1

### Supplemental Information
Supplemental information for this article can be found online at http://dx.doi.org/10.7717/peerj.16788#supplemental-information.

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
