# Peer review of "Predicting repeat power ability through common field assessments: is repeat power ability a unique physical quality?"

_PeerJ, doi:10.7717/peerj.16788_

## Round 0.1 · original submission · Major Revisions

The reviewers find merit in the manuscript. However, some specific points need further clarification, such as the main purpose, and more information is needed about methods and procedures to allow replication. Please, address each comment point by point and provide a detailed response.

Reviewer 1 ·

Basic reporting

Please see the pdf attached.

Experimental design

Please see the pdf attached.

Validity of the findings

Please see the pdf attached.

Additional comments

Please see the pdf attached.

Annotated reviews are not available for download in order to protect the identity of reviewers who chose to remain anonymous.

Reviewer 2 ·

Basic reporting

No comment.

Experimental design

No comment.

Validity of the findings

No comment.

Additional comments

Dear authors,

Very nice paper about the obtaining more information the neuromuscular and metabolic qualities of Repeat Power Ability. Throughout the entire manuscript, there is sufficient scientific evidence that supports the claims and the findings of the authors. In this way, the authors should be congratulated for the well-written manuscript and the well-conducted research.
However, there are some specific changes and suggestions that should be made to improve the quality of the paper.

Introduction:
- Pg8Ln58 - This sentence needs a reference.
- What were the hypotheses of the study? Insert this information in the introduction section
after the objective.

Materials & Methods
Participants
- What were the inclusion and exclusion criteria for participants. Please include this information.

Procedures
- Was any familiarization carried out with the participants? If yes, enter this information. If no familiarization was carried out, the results obtained in the study may have been compromised, since the improvements found may be due to the fact that the participants improved their technical level in each exercise/test.
- Were the tests performed at the same time in all four sessions? Please include this information.

Coutermovement Jump
- Enter the ICC and CV percentage
- Pg12Ln168 - Enter the name of the force platform, model, country and city.

Speed
- Enter the ICC and CV percentage
- In the warm-up, how did the authors control the intensity of 70, 80 and 90% in the sprints? Enter this information.

Yoyo Intermittent Recovery Test 2
- Enter the ICC and CV percentage

Statistical Analysis
- Authors in the statistical analysis section must mention the level of significance used.
- Pg17Ln278 - Enter the name of the model, country and city of the software

Results
- Authors must place a caption in tables 1, 2 and 3 describing the meaning of the acronyms.
- The authors refer in the results to a table 4 that is not included in the annexes. Rectify this
situation.
- The authors in the annexes have a table 3 that they never refer to throughout the document.
Please rectify

Discussion
- The first paragraph of the discussion should be a reminder of the objective of the study. Please include this information.
- Pg21Ln381 - This sentence needs a reference.
- It would be interesting to place a paragraph after the limitations mentioning future lines of investigation, in order to enrich the study.

Best regards

---

## Round 0.2 · accepted · Accept

The authors have addressed all of the reviewers' comments and the manuscript is ready for publication.

Reviewer 1 ·

Basic reporting

No comment

Experimental design

No comment

Validity of the findings

No comment

Additional comments

The authors did a great job in reviewing the manuscript as requested. I have no further comments. Congratulations on a very interesting study. All the best.

Reviewer 2 ·

Basic reporting

no comment

Experimental design

no comment

Validity of the findings

no comment

Additional comments

no comment